# Rebound Pain After Regional Anaesthesia

**DOI:** 10.3390/medicina61050790

**Published:** 2025-04-24

**Authors:** Frances Fallon, Mohd Shazrul Ramly, Aneurin Moorthy

**Affiliations:** 1Department of Anaesthesia, Children’s Health Ireland at Temple Street, D01 XD99 Dublin, Ireland; 2Department of Anaesthesia, Mater Misericordiae University Hospital, D07 R2WY Dublin, Ireland; 3School of Medicine, University College Dublin, D04 W6F6 Dublin, Ireland; 4Department of Anaesthesia, National Orthopaedic Hospital Cappagh, D11 EV29 Dublin, Ireland; 5The ESA-IC Oncoanaesthesiology Research Group and Outcomes Research, Cleveland, OH 44195, USA

**Keywords:** rebound pain, regional anaesthesia, acute pain, surgery

## Abstract

The last decade of anaesthesia practice worldwide has seen considerable advancements in the field of regional anaesthesia with new equipment, techniques, and drug developments. With these advancements, regional anaesthesia practice has gained considerable momentum, and more patients benefit from it. Here, we review rebound pain after regional anaesthesia, a common yet poorly understood phenomenon that all regional anaesthesiologists should be familiar with in order to recognise, manage, and, where possible, prevent it.

## 1. Introduction

Since the advent of ultrasound-guided regional anaesthesia (RA) over the past two decades in anaesthesiology practice, rebound pain (RP) has become a clinical phenomenon of concern [1,2]. The major clinical characteristic of rebound pain is an abrupt, severe pain that occurs after the resolution of a peripheral nerve block (PNB) and can be severe enough for patients to seek medical attention. This poses an important challenge to anaesthesiologists and their patients, with wider implications for general practitioners and emergency medicine specialists. Recent studies have focused on identifying mechanisms to predict and prevent RP. While some success has been found in the latter pursuit using dexamethasone, attempts to create valid risk prediction models have, thus, been unsuccessful. Regional anaesthesia is a core element of many multimodal analgesia techniques that are particularly important in the context of ambulatory surgery and account for over half of the surgical procedures in the developed world [3,4]. The benefits of RA are well established and lend themselves to an ambulatory setting: high-quality analgesia, reduced post-operative nausea and vomiting, reduced nosocomial infection rates, faster discharge from post-anaesthesia care units, reduced morbidity and mortality, and increased patient satisfaction [3,5,6]. However, RP presents a clinically relevant problem and may limit the utility of PNBs in certain patients where it may occur outside a controlled healthcare setting [7]. In this review, we outline the understanding of RP and the difficulties it poses for those who employ RA as part of their practice. We will also outline the risk factors for RP that may be identified perioperatively and the methods that can be employed to prevent it in everyday clinical practice.

## 2. Defining Rebound Pain and Its Prevalence

There are various definitions of RP in the literature. Among the several existing definitions of RP, the single unifying feature is that it occurs after the resolution of a nerve block. Beyond this, there are numerous descriptions of the onset timing, quantifying pain, and the mechanism by which it occurs. One of the earliest definitions in the literature outlines RP as a quantifiable difference in pain after peri-neural local anaesthetic wears off [8]. A more recent definition specifies quantifiable parameters as a change in the numerical rating scale from mild (≤3) to severe pain (≥7) within 24 h of block performance [9]. Other definitions refer to mechanical–surgical pain and hyperalgesia [4,10,11]. A recent, more holistic definition refers not only to the presence of acute severe pain following the resolution of the sensory block but also to the impact of pain on psychological well-being, quality of recovery, and activities of daily living [12]. Such impacts are probably underappreciated in the literature but have significant implications for everyday practice and patient satisfaction.

The prevalence of RP has been reported to be between 35% and 49.6% [9,13,14]. As such, it is common yet perhaps under-recognised in clinical practice, especially in an ambulatory surgery setting where patients initially go home pain-free. However, RP is associated with the need for unplanned medical input, with one study reporting that 9.8% of patients required unplanned healthcare practitioner input after a popliteal nerve block [15]. Retrospective data from 2016 showed similar results, where 12% of patients receiving single-shot brachial plexus block had an unplanned physician visit versus 4% of patients who underwent general anaesthesia alone [13].

## 3. Recognising Rebound Pain and Understanding Its Mechanisms

Rebound pain can be recognised based on the existing definitions and well-defined features (Table 1). In all cases, it is a transient phenomenon [13]. The timing of onset is consistent with the offset of sensory blockade, which typically coincides with nighttime pain [4,16]. It is typically described as a burning pain, most commonly lasting from 2 to 6 h [12,17,18]. Less frequently, it is described as a dull ache [18]. The pain is characteristically severe and is measured by a numerical rating score of ≥7 within 24 h of block administration [9].

The exact pathophysiology of RP remains to be fully elucidated and agreed upon. Figure 1 summarises current hypotheses on the mechanism of rebound pain. Often, the argument posed is one of block wearing-off versus a neuroinflammation-induced hyperalgesia mechanism of the block itself. Needle insertion and pressure trauma have also been suggested to be contributing factors [4]. With respect to the neuroinflammatory theory, bupivacaine has been shown to cause transient neurotoxicity via Schwann cell degeneration and demyelination [11]. The occurrence of hyperalgesia has been supported by animal studies [11,20,21]. In one such study, when ropivacaine sciatic nerve blocks in rats were resolved, transient hyperalgesia to thermal stimuli was observed compared to placebo [11]. However, it is argued that this neuroinflammatory hypothesis does not fully explain the short duration of RP [12]. Furthermore, hyperalgesia is a well-established feature of tissue injury after surgical trauma that occurs in the absence of RA [12,22].

A particularly popular hypothesis is the unmasking of nociceptive response to surgery [12]. This unopposed nociceptive input that occurs when neural blockade subsides is believed to be due to mechanical–surgical pain. The pain felt is that which would be expected rather than an exaggerated response. The ‘masking’ prior to this is understood to be due to the effects of local anaesthetic blockade, which limits signal transduction from peripheral nerves to second-order neurons in the spinal cord. Subsequent transmission via the spinothalamic pathway to the brain is limited [7,12].

## 4. The Challenge Posed by Rebound Pain—Is It Worth the Risk?

There are considerable concerns regarding patient dissatisfaction with RP after RA. The negative effect of RP on patient experience should not be overlooked [23]. There remains a lack of prospective clinical studies examining the degree of dissatisfaction and its impact on patient experience, and this remains an important future research question. Recently, patient satisfaction scales (e.g., quality of recovery score) have been utilised as the primary outcome in numerous randomised controlled trials comparing the efficacy of the two RA techniques and including a placebo arm [24,25,26]. Future studies investigating RP should include patient satisfaction scales such as the quality of recovery score (QoR-15) instead of using a one-dimensional pain scale to reflect the patient’s post-operative experience.

One retrospective study reported that 11% of patients who underwent PNB were not satisfied with the procedure. In the same study, 52% of the patients had RP, and RP was the most commonly cited factor leading to dissatisfaction. Interestingly, only 24% of those who were not satisfied said that they would be unwilling to undergo the same RA procedure again [23]. Our understanding of RP is not complete with these findings because not all participants with RP (which, by most definitions, is severe) were dissatisfied, suggesting that there may be other factors at play. For example, patient dissatisfaction with RA has been associated with pain during the procedure [27]. More recently, in the literature, it was found that despite 49.6% of a cohort of 972 patients with RP, overall patient satisfaction rates were 83.2%, again suggesting that there is more to patient satisfaction than RP. Factors such as quality of care, length of stay, and staff engagement may also play a role, and further investigation is warranted [9].

The challenges posed by RP are not limited to patient satisfaction alone, with potential increases in healthcare workload and costs suggested, but are largely under-studied [28]. Another concern that warrants further investigation is the contribution of acute post-operative pain in the form of RP to pulmonary and cardiovascular complications [29].

## 5. Risk Factors

Can RP be predicted using a risk-prediction model? One group developed a risk prediction model for severe RP after foot and ankle surgery involving a single-shot popliteal sciatic nerve block. However, in this retrospective cohort study (*n* = 1365), the multivariate risk prediction model showed poor predictive performance for severe RP [15]. The authors found that the event rate of RP was 49.7%, with 9.8% requiring unplanned healthcare practitioner contact. Several factors were noted to be important variables: female sex, significant bony surgery involving the foot and ankle, tourniquet time, and the use of dexamethasone. Other variables that were collected in this study were intraoperative anaesthesia type, age, and admission vs. ambulatory surgery. The authors concluded that routinely collected clinical data are not adequate to provide individualised risk stratification for RP. It was suggested that patient-centred predictors such as severity of acute or chronic pain, pre-operative opioid use, anxiety, and depression may play a role in future studies to improve prediction model performance.

To date, several studies have identified modifiable and non-modifiable risk factors associated with RP after RA. The modifiable risk factors include the use of an indwelling catheter with local anaesthetic, adjuvant agents (e.g., perineural dexamethasone with local anaesthetic agents or intravenous dexamethasone), and blocking of regions with dense nerve distribution, such as the brachial plexus and popliteal space [12,30]. Another modifiable risk factor is patient education and management of expectations. Falsely low pain tolerance may be associated with fear of pain, nerve damage, or poor understanding of the course of PNB [18].

Non-modifiable risk factors include female sex, young age, and surgery involving the bones. Figure 2 summarises the non-modifiable and modifiable risk factors. In a systematic review of 50 studies, young age, female sex, and those undergoing orthopaedic surgical procedures, especially upper extremity surgeries, were at risk of RP. The same review also identified that RP was associated with patients’ cognitive functioning and anticipation of post-operative pain [30]. A single-centre retrospective study (*n* = 1177) reported that age < 55 years, Indian ethnicity, and surgeries involving the shoulder, tibia, or below-knee amputation were associated with higher RP scores [31]. Of note, RP is 6.5 times more likely to occur in bone than in soft tissue surgery and is independently associated with single-shot PNB [30]. There is a higher pain prevalence in the female gender. However, the reason is not fully understood. This may relate to a complex interplay between physiological (hormonal differences and variations), psychological, and social factors (such as coping skills, social roles) [32,33]. With regard to age as a non-modifiable risk factor for RP, it is suggested that younger individuals might have a different pain perception, potentially causing a heightened sensitivity to pain.

## 6. Prevention: Better than Cure, but Is It Possible?

Several strategies have been suggested for preventing RP after RA (Figure 3). These include routine prescribing of multimodal analgesia and its timely initiation, the use of continuous infusion of local anaesthetic agents via catheter technique, the use of adjuncts with the local anaesthetic, and patient education with management of expectations [12].

Multimodal anaesthesia typically involves several pharmacological agents, including opioids, NSAIDs, paracetamol, and gabapentin, in conjunction with the use of LA itself. Gabapentin has been evaluated extensively in the literature for its role in the management of acute post-operative pain for a number of different procedures, and its use as a ‘protective premedication’ has been explored [34]. Specifically with respect to minor and major oncological breast surgery, its use pre-operatively is recommended in PROSPECT guidelines [35]. This is due to its demonstrated success at reducing post-operative opioid consumption and reduced pain scores immediately post-operatively and at 24 h [36,37]. PROSPECT also suggest that it can be considered pre-operatively for abdominal hysterectomy due to numerous encouraging studies demonstrating improved patient satisfaction, post-operative pain scores, and lower opioid usage post-op [38]. Gabapentin is antihyperalgesic and is understood to reduce the hyperexcitability of dorsal horn neurons induced by tissue damage [34]. It is used post-operatively for acute pain and has been suggested as a rescue for rebound pain; however, this has yet to be thoroughly investigated [8,39].

The use of dexamethasone has been extensively studied in the context of RP prevention (Table 2). Two RCTs on the use of perineural dexamethasone as an adjunct to PNB showed a lower incidence of RP (Woo et al.: 37.1% vs. 81.9%, *p* < 0.001 and Fang et al.: 11.1% vs. 48.8%, *p* = 0.001), with the latter also showing reduced 24-hour opioid consumption [17,40]. Another small RCT (*n* = 72) found that perineural dexamethasone prolonged block duration. However, this study found an increase in RP and RP scores [41]. A network meta-analysis on dexamethasone use found that the use of both intravenous and perineural dexamethasone reduced the incidence of RP, with intravenous dexamethasone ranked first on the *p*-score. Secondary outcomes included time to first analgesic request, RP resolution time, the difference in pain score before and after PNB, and nausea/vomiting, all of which had a favourable response to perineural and IV dexamethasone [42]. Another meta-analysis, which included seven RCTs (*n* = 574), also found that dexamethasone (either intravenous or perineural) was effective in reducing rebound pain after a single injection nerve block [43]. The same authors followed up this analysis recently to conclude via meta-analysis that IV dexamethasone was effective, while perineural dexamethasone was likely to have some benefit [44]. Given that perineural dexamethasone is an off-label form of delivery, an IV route is preferable in everyday clinical practice. These suggested benefits from prospective trials in the future for robust evaluation.

RP may also be prevented using an indwelling local anaesthetic infusion catheter in patients undergoing RA. One RCT (*n* = 92) found that the single-shot technique with IV dexamethasone had a higher incidence of RP than continuous infusion vs. interscalene indwelling catheter (65% vs. 20%, *p* = 0.01), with the catheter group having lower opioid consumption at 24 h post-operatively [46]. Similarly, a patient-controlled indwelling catheter analgesia (PCIA) group had reduced post-operative pain in the first 24 h, with no RP reported when compared with both the single-shot interscalene brachial plexus blockade and control groups in a three-armed RCT (*n* = 154) [47].

The use of NMDA receptor antagonists has also been examined to reduce the risk of RP after PNB. One study did not find that the use of 0.3 mg/kg/h ketamine intraoperatively reduced RP intensity or incidence [48]. Conversely, another study found that the use of esketamine after thoracoscopic lobectomy with paravertebral block reduced the incidence of RP at day 7 post-operatively (8.3% vs. 33.3%, *p* < 0.001) with a lower mean pain score at 24 and 48 h in the esketamine group (*p* = 0.09 and *p* = 0.03, respectively) [49]. The discord in findings is difficult to elucidate given the variance in the drug used. Further studies could compare ketamine vs. esketamine to determine if a true benefit lies with either, with the interrogation of the effects of different doses. The use of perineural ketamine has also been investigated for its role in the prevention of rebound pain, and one RCT found that it inhibited rebound pain, while IV ketamine failed to have any effect [50]. A recent meta-analysis of 12 RCTs was also found to be in favour of perineural ketamine with respect to block duration, although RP was not an endpoint in this analysis, and further investigation is warranted [51].

Optimising patient perceptions and expectations is an important aspect of RP mitigation [4,30]. Education is a modifiable risk factor that can be addressed in several ways. Some studies have shown that written materials improved patients’ understanding of PNB [52]. Other successful methods of educating patients undergoing anaesthesia include videotape and CBT; however, their use specifically in RP has not been investigated [53,54]. Thorough and repeated clinical information for patients may increase the overall utility of PNBs and improve patient experience.

Novel non-pharmacological approaches for the prevention of rebound pain are also being investigated. One example is neuromodulation via vagal nerve stimulation, which targets the nociceptive hyperexcitability implicated in rebound pain. Vagal nerve stimulation has also been proposed to relieve acute pain [55]. One RCT found that transauricular vagus nerve stimulation resulted in a significantly lower incidence of rebound pain compared to a sham stimulation group in patients undergoing anterior cruciate ligament repair with pre-operative single-shot femoral nerve block [56]. Larger RCTs are required in this area, and as our understanding of the mechanisms of rebound pain expands, it is likely that the variety of novel approaches to tackle it will too.

## 7. Conclusions

Rebound pain remains a clinical challenge. Anaesthesiologists should be aware of this important phenomenon when performing RA and tailor their multimodal analgesia strategy to each individual patient with the goal of minimising the risk of RP during the post-operative period. There is a paucity of robust, large-scale RCTs on the matter, but momentum is growing in the literature. Thus far, our understanding is limited, and a definition has yet to be agreed upon. Further studies and research should guide our understanding of the pathophysiology of RP, aid in identifying at-risk patients, and develop further prevention strategies. In the meantime, anaesthetists should be mindful of RP, the risk factors, and potential approaches to reduce its occurrence. While unanswered questions regarding RP currently pose a challenge in clinical practice, they also signal potential research agendas in the near future.

## Figures and Tables

**Figure 1 medicina-61-00790-f001:**
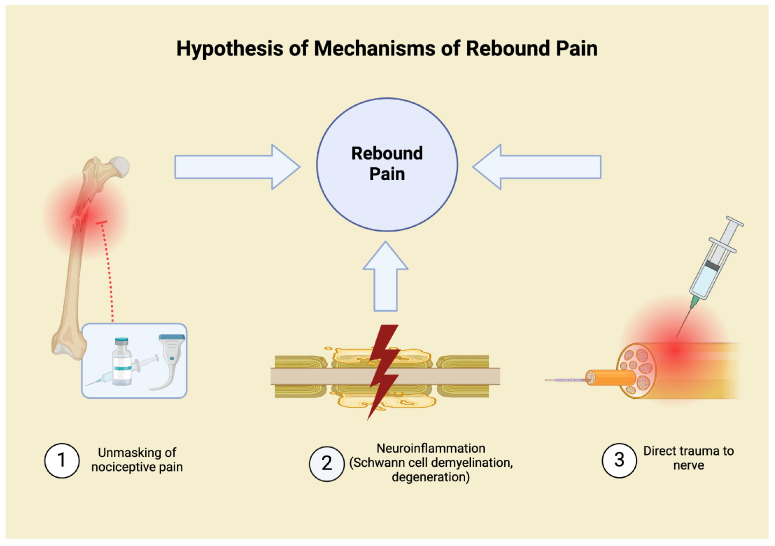
Some of the current hypotheses on the mechanisms of rebound pain. Unmasking of nociceptive pain, direct nerve injury, and neuroinflammation have all been proposed. The complete mechanism has yet to be fully elucidated.

**Figure 2 medicina-61-00790-f002:**
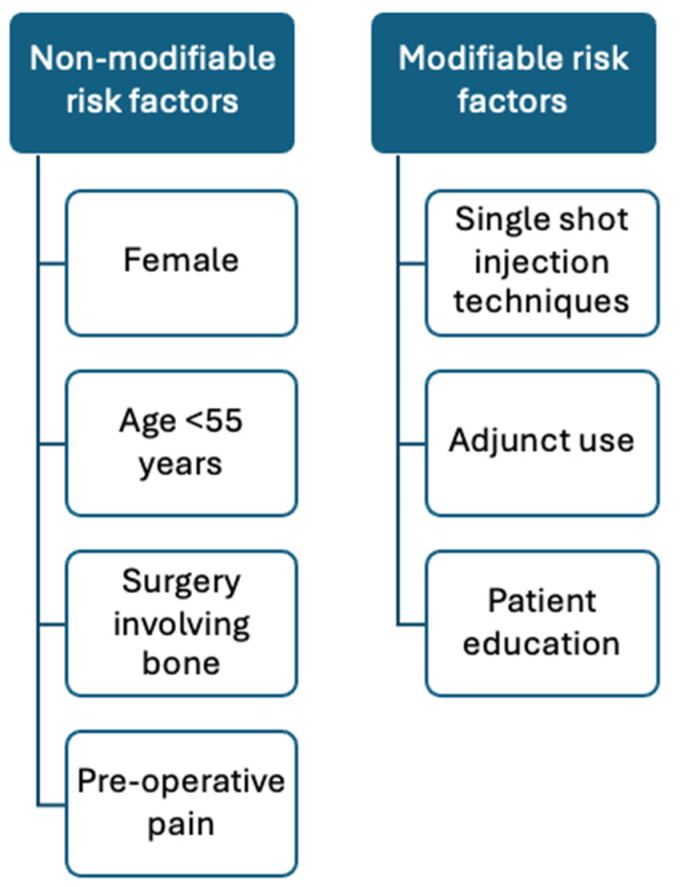
Risk factors for rebound pain.

**Figure 3 medicina-61-00790-f003:**
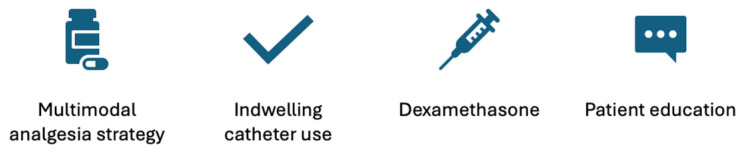
Methods of prevention of rebound pain.

**Table 1 medicina-61-00790-t001:** Features of rebound pain [8,18,19].

Characteristics	Description
Onset	Following resolution of peripheral nerve blockade
Timing	Typically at night
Severity	≥7 on numerical rating scale
Quality	Typically burning or dull
Occurrence	At rest or movement
Duration	Typically 2 h
Neuropathic features	Absent

**Table 2 medicina-61-00790-t002:** Current evidence on intravenous and perineural dexamethasone.

**Dexamethasone: Intravenous and Perineural [40,41,43,45]**
Reduce RP incidence Longer time to first analgesic request Improved pain score after PNB Reduced nausea/vomiting post-operative
**Dexamethasone: Perineural Only**
Reduced 24-hour opioid consumption Prolonged block duration

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
