# Peer review of "Rebound Pain After Regional Anaesthesia"

_medicina, 2025, doi:10.3390/medicina61050790_

Round 1
Reviewer 1 Report
Comments and Suggestions for Authors
A review of the literature on medical procedures for rebound pain after the use of local anaesthesia techniques was conducted. The lack of information on the method of selecting articles for the review is noteworthy.
Please provide the criteria for selecting source materials that were taken into account during the review, in particular the period of publication, key words, sample size, tools for selecting literature for the review.
The manuscript submitted for review should be supplemented with several issues:
Line 90-98. Please clarify the information on local anaesthetics from the amide group, whether hyperagesia was observed in the case of these two drugs or others from this chemical group?
Line 100-105. Please explain the thesis. During anaesthesia, the patient feels signals from the area of ​​the procedure, i.e. mechanical-traumatic stimuli, to a limited extent. However, after the blockade induced by local anaesthesia has ceased, is the subsequent transmission limited or intensified? Based on this paragraph, it is not sufficiently clear.
Lines 122-133. What are the authors' assumptions regarding the reasons for this discrepancy - satisfaction or lack thereof in relation to the decision to undergo the procedure again? Were the scale of pain therapy used during the convalescence period, the quality of medical care and staff involvement taken into account?
Lines 141-143. Please discuss this model in more detail, what were its shortcomings and what factors were taken into account when creating the model predicting the risk of a severe course?
Lines 171-180. Multimodal pain control techniques also include groups of drugs other than the cited ketamine, opioids and local anesthetics. Were drugs from the GABA analogue group, antidepressants, anticonvulsants and anxiolytics included, which, in addition to their effect in multidrug therapy, can modify the patient's mental state, including fear before the procedure and then fear of expected pain? What significance can non-opioid analgesics and non-steroidal anti-inflammatory drugs have? In the cited works, were patients given a choice of preparations using additive synergism of peripherally and centrally acting agents?
Line 215 – 219. The discussion should be concluded with a summary, citing conclusions and more detailed suggestions appearing in the text.
In the introduction, the authors announce a discussion of RP risk factors and methods (lines 37-39) that can be used to prevent this phenomenon, which should be implemented, which was not done exhaustively.
Author Response
See attached response.
Many thanks

Reviewer 2 Report
Comments and Suggestions for Authors
In the article, an examination of rebound pain after regional anesthesia is presented.
Citation formats should be arranged within the article.
It would be appropriate to add a paragraph summarizing the contributions made to the literature at the end of the Introduction section.
Table titles should be on the top row of the tables.
Why are gender and age important as non-modifiable risk factors? Why are these criteria related to pain?
Author Response
See attached response.
Many thanks

Reviewer 3 Report
Comments and Suggestions for Authors
The manuscript is well-written and informative but would benefit from minor structural and content enhancements to maximize its impact.
Suggestions for improvement: Consider adding a concise summary table or flowchart to visually depict the proposed mechanisms (neuroinflammation, unmasking nociception, etc.) for better reader comprehension. The section on dexamethasone and adjuncts is thorough, but a comparative table (e.g., efficacy of IV vs. perineural dexamethasone) would enhance clarity. Expand on specific educational interventions (e.g., pre-operative counseling materials, digital tools) that have shown efficacy in managing expectations. Highlight gaps more explicitly—e.g., the need for standardized RP definitions, multicenter RCTs on prevention, or long-term patient-reported outcomes. Briefly discuss how anesthesiologists can integrate these findings into practice (e.g., risk stratification protocols).
Author Response
See attached response
Many thanks

Round 2
Reviewer 1 Report
Comments and Suggestions for Authors
I would like to thank the authors for improving the manuscript and clarifying all doubts. The article is ready for publication and is very much needed to raise awareness and sensitize the community to the patient's discomfort in the postoperative period and during recovery. I recommend publishing the manuscript in its current form.